# Determination of Load-Carrying Capacity of C-Profile Glued Ti-Al Column under Temperature Environment

**DOI:** 10.3390/ma14113013

**Published:** 2021-06-02

**Authors:** Leszek Czechowski

**Affiliations:** Department of Strength of Materials, Lodz University of Technology, 90-924 Lodz, Poland; leszek.czechowski@p.lodz.pl; Tel.:+48-42-631-22-15

**Keywords:** thermal analysis, finite element method, glued structures, experimental study

## Abstract

The paper deals with an examination of the behaviour of glued Ti-Al column under compression at elevated temperature. The tests of compressed columns with initial load were performed at different temperatures to obtain their characteristics and the load-carrying capacity. The deformations of columns during tests were registered by employing non-contact Digital Image Correlation Aramis^®^ System. The numerical computations based on finite element method by using two different discrete models were carried out to validate the empirical results. To solve the problems, true stress-logarithmic strain curves of one-directional tensile tests dependent on temperature both for considered metals and glue were implemented to software. Numerical estimations based on Green–Lagrange equations for large deflections and strains were conducted. The paper reveals the influence of temperature on the behaviour of compressed C-profile Ti-Al columns. It was verified how the load-carrying capacity of glued bi-metal column decreases with an increase in the temperature increment. The achieved maximum loads at temperature 200 °C dropped by 2.5 times related to maximum loads at ambient temperature.

## 1. Introduction

The analysis of the load-carrying capacity of thin-walled structures is still a challenge for engineers with regard to many appearing phenomena. This leads to a necessity of individual approach if different materials and the combination of them are considered. Moreover, different boundary conditions of structure work can essentially influence the final results of the buckling load and the load-carrying capacity. Untypical condition of the work can be treated as the thermal environment, among others. This is caused by the fact that properties of materials can change considerably with a change of temperature. Lately, this phenomenon at analysis of the structures strength is more often taken into account. The analysis of the strength of metal beams in higher temperature was performed in papers [1,2]. Authors of other works examined steel samples at elevated temperature to determine their material properties [3,4,5]. Nguyena et al. [6] analysed the behaviour of the carbon fibre-reinforced polymer (CFRP) structures due to temperature environment action and different mechanical loads. Jin et al. in their papers [7,8] investigated the thermal buckling of circular aluminium and laminated composite plates by applying the Digital Image Correlation (DIC) method. Authors of paper [9] examined experimentally metallic structures under mechanical loads at higher temperatures and in accident conditions. They stated that the temperature with some materials can play huge role on the behaviour of structures. Authors of papers [10,11] studied the load-carrying capacity of compressed thin-walled steel and titanium C-column at temperature up to 200 °C. Zhou et al. in paper [12] analysed mechanical properties of CFRP composites at elevated temperature. Authors of work [13] explored the behaviour of GFRP profiles due to compression at elevated temperature. Zhang et al. in [14] determined both at ambient and at elevated temperature the mechanical properties of IN718 alloy produced by laser metal deposition. The work [15] presents the analysis of thermal buckling of clamped panels under global or local heating. This paper includes the results of some nonlinear behaviour of a structure registered by a three-dimensional DIC system. Wang et el. in papers [16,17] analysed the buckling and the creep buckling of steel columns subjected to high temperature according to fire resistant standard ISO-834. Wong and Wang in [18] investigated experimentally the behaviour of the pultruded glass fibre-reinforced plastics (GFRP) channel columns at elevated temperatures. One the other hand, in accessible literature one can find many works relating to thermal buckling analysis of composite or functionally graded material structures based on numerical or analytical calculations in papers [19,20,21,22,23,24,25,26], among others. Authors of paper [27] examined the strength of cold-formed lipped channel beams due to elevated temperatures. To assess the ultimate strength for higher temperatures they used Direct Strength Method (DSM). In the literature there are many papers which involve the analysis of the buckling and the post-buckling state of the thin-walled structures. Experimental and numerical studies on structures built of different material are widely described in papers [28,29,30,31,32,33,34,35,36,37,38,39,40,41,42,43]. Authors of these papers focused on the strength and the stability of structures. The present paper also links to mentioned works because the stability of thin-walled structures was analysed as well but in this case under thermal load. Moreover, the considered columns were made of two metals (titanium and aluminium) connected to each other by using temperature-resistant adhesive. Krahmer et al. in paper [44] dealt with the determining effects on surface state and tensile test with respect to manufacturing method. Based on low-carbon steel and Inconel 718, they showed the influence of the removal process on the surface integrity and the surface state. Owing to this fact, border effects can play a significant role in obtained results.

By taking a look at the literature, the present study is novel as the stability of glued profiles as FML (Fiber Metal Laminate) was analysed in [39,40,42], among others, but those columns were usually examined under mechanical loads. Besides, the techniques and materials of the columns fabrications in the present case were completely different. Thus, the assessment of the behaviour of glued structures made of different metals is still desirable, especially at higher temperature. Therefore, the present analysis evaluates the behaviour of Ti-Al glued C-profile under compression at elevated temperature. The shape of the grooves of compression plates acting on the columns edges during tests was elaborated to reflect articulated supports. The examination of the strength of columns was performed until the load-carrying capacity was achieved. The tests in higher temperatures were continued even if some (local) separation between metals occurred. While the experiment was conducted, non-contact Digital Image Correlation Aramis^®^ System (DICAS) [45] destined to record deformations in samples was employed. Applied Aramis^®^ System for the maps registration was based on commercial GOM Correlate software. To validate the empirical results, full characteristics of titanium and aluminium dependent on temperature were taken into consideration. In the case of applied adhesive, materials properties were based on the product card by assuming bilinear characteristics in a function of temperature with increment every 25 °C. It was assessed how the load-carrying capacities of Ti-Al columns due to compression decrease with an increase in temperature. Moreover, the strengths values of considered metals were different and this fact can substantially affect the columns behaviour of columns. It was observed that deformations of columns changed with the temperature of the test with respect to possible separations of metals or a change of the properties.

## 2. Problem Description

### 2.1. The Study Object

Ti-Al column at length of 250 mm (L), mid-dimensions: width 80 mm (b), and height 40 mm (a) and total thickness 1.2 mm (t_t_) subjected to compression at elevated temperature was examined (Figure 1). The thickness of aluminum wall and titanium wall was equal to t_Al_ = 0.5 mm and t_Ti_ = 0.5 mm, respectively. The columns were built of titanium (type: grade 2, content of elements: T > 98,9%, Fe < 0.3%, O < 0.25%, C < 0.08%) and of aluminum (type EN AW-5754 H22, content of elements: Ti ≤ 0.15%, Mg = 2.6–3.6%, Mn ≤ 0.5%, Fe ≤ 0.4%, Si ≤ 0.4%, Cu ≤ 0.1%, Zn ≤ 0.2%, Cr ≤ 0.3%, remaining part contributed aluminum) glued by structural adhesive Araldite^®^ AW 4804/Hardener HW 4804 delivered by Huntsman company (Basel, Switzerland). The weight parts of the glue components were the following: AW 4804—87% and HW 4804—13%. With accord to a producer offer [46], using aluminium-filled epoxy adhesive ensures the connection between glued metals at a temperature coming up to 210 °C, at least. Moreover, this adhesive is characterised by high strength, moderately high stiffness, high creep and fatigue resistance at low shrinkage. The compression of the columns at elevated temperature took place in a thermal chamber (Instron, model 3119—605, Norwood, MA, USA) using an Instron machine—model 5982 (Figure 2). On this strength machine one can test the samples with force within 0.02 N to 100 kN, whereas using the thermal chamber enables work at temperatures up to 350 °C. To follow the deformation during tests, the samples were covered with the high-temperature-resistant paint coat.

The experiment was composed of three stages. In the first stage, columns were compressed with initial load (circa 5–10% of maximum load obtained at ambient temperature). The second stage related to temperature increase to an appropriate temperature while the grips of machine were unmovable which could increase the reaction forces. The process of warming up the samples lasted about 1–2 h, at most. The adequate compression of the column at elevated temperature involved the last stage.

### 2.2. Material Data

Based on the one-directional tensile tests performed at ambient temperature, 100 °C, 150 °C and 200 °C, full material characteristics for titanium (Figure 3a) and aluminum (Figure 4a) were obtained. Due to linear interpolation, characteristics for other temperatures were also determined. This enabled greater gradations of mechanical properties dependent on temperature. Finally, all data were transformed to curves: true stresses vs. logarithmic strains and implemented to software (Figure 3b and Figure 4b). Material data of adhesive were assumed based on the product card [46]. 

In case of adhesive, the mean Young’s modulus was assumed to be dependent on temperature according to Figure 5 (for ambient temperature E_glue_ = 6.2 GPa). Young’s modulus for ambient temperature for titanium and aluminum alloy was 107 GPa and 70 GPa, respectively. Poisson’s ratio was temperature-independent (0.28 for titanium and 0.33 for aluminum). The thermal expansion coefficients of titanium and aluminum were determined by measuring the strains in a range from 23 °C to 200 °C (α_Ti_ = 8.8^−6^ 1/K, α_Al_ = 23.1^−6^ 1/K). Thermal expansion coefficients of glue were based on product card (α_glue_ = 50^−6^ 1/K). Consequently, this value was taken into account.

### 2.3. FE Model 1

Numerical simulations were done based on the finite element method (FEM). The first numerical model (denoted as FEM_SOLID) based on the solid elements was elaborated in NASTRAN/MARC FEA 2010R^®^ version software [47]. A discrete model built of solid elements was created by extruding the first-order shell finite elements (Figure 6). The number of elements across the thickness of column was assumed to be 5. Consequently, the number of solid elements came to about 150,000. However, the total number of all elements (including elements of other parts) amounted to above 300,000. 

Numerical computations were done by switching on Green–Lagrange equations for large strains and deflections and the second Piola–Kirchhoff stress as well [47]. The number of sub-steps for each performed simulation was between 500 and 1,000,000. The iteration number for each sub-step was assumed to be from 10 up to 10,000. The setting of boundary conditions and the stages related to similar conditions of the experiment. The load-step-1 denoted as LS-1 was based on the small shortening (U_y_) of the columns. This allowed closing the gaps between discrete models (plates and column). The friction coefficient between all parts amounted to 0.05. The contact detection parameter was based on bias on tolerance. The separation criterion of forces was considered in calculations. The load-step-2 denoted as LS-2 involved an increase in temperature by some temperature increment. Then, both plates (upper and lower) holding the column on their external edges were unmovable (U_x_ = U_y_ = U_z_ = 0). The last load-step-3 (LS-3) related to adequate compression of column at elevated temperature. The relations between the column edges and supporting plates are illustrated in Figure 7. In all simulations, initial imperfections of columns were omitted.

### 2.4. FE Model 2

The second numerical models considered in the present analysis were based on shell elements. The three stages descripted in Section 2.3 (LS-1, LS-2, LS-3) were realised as well but discrete shell models were supported on outer nodes (on edges). To compare, two shell models with different boundary conditions were taken into consideration (denoted as FEM_SHELL_1 and FEM_SHELL_2). 

The first of them assumes the master nodes connected with nodes as being on the outer edges of the column (FEM_SHELL_1—Figure 8). It means that the support and the displacement of column followed due to two operating nodes. Moreover, FEM_SHELL_1 enabled gentle eccentricity (given some displacement e_z_) of the gravity of the center of the column because the aluminium and titanium possess slightly different mechanical properties and this could influence the force distribution on the edges during compression and different behaviour of the column. The second one related to articulated supports on the edges of the column (FEM_SHELL_2—Figure 9). Besides, the meshes of the models were identical. The calculations were carried out by using the 18.2 ANSYS^®^ software version [48] by means of Shell281 type. This element is based on eight nodes at six degrees of freedom. Moreover, the applied element can be used both in linear and non-linear analysis of multilayers structures with material nonlinearity. The accuracy of modelling the multilayer materials relies on the first-order deformation theory (Mindlin-Reissner’s theory). The finite element in the considered analyses was assumed to be 2 mm. Then, total number of shell elements in the discrete model was 10,000. In simulation, isotropic hardening of materials in elastic-plastic range was considered. As previous model, discrete models were treated as the perfect ones.

## 3. Results

### 3.1. Compression at Ambient Temperature

Figure 10 illustrates the curves of compression (reaction) force F_R_ vs. shortening Δ for all considered methods. Based on the results, it can be noticed that experiment curves reach considerably greater shortening at maximum loads than in case of FEM results (except FEM_SHELL_2). It means that in first phase of compression, the adjustment of columns on supports could follow. If one can consider the shift of curve by some value (denoted as FEM_SOLID SHIFT), then the curve is very close to experimental curves (obtained stiffness is comparable). The FEM_SHELL_2 (at articulated supports of column) gave significantly higher maximum load (F_R_ = 21.66 kN) in a comparison to experiment (F_R_ = 15.33 kN). Moreover, in this case maximum load has been achieved just at shortening circa 1.5 mm. The slight eccentricity of applied force by e = 1 mm, e = 2 mm and e = 3 mm caused an light increase in the load-carrying capacity. However, curves of FEM_SHELL_1 for e = 1 mm and FEM_SHELL_1 for e = 2 mm are comparable with the curve based on FEM_SOLID calculation as well, and the maximum loads are also close (See the maximum loads set out in Table 1).

The maps of deformations for four different loads are shown in Figure 11. First points (I) and (II) denote the shortening of 0.2 mm and 0.4 mm, respectively. The point (III) represents the deformations of column at maximum load F_Rmax_ (for an adequate case). The point (IV) refers to the deformation of columns at the end of experiment or calculations. By relating to experimental maps, cases FEM_SOLID, FEM_SHELL_1 with e = 1 mm and FEM_SHELL_2 are very similar, especially for shortening 0.4 mm. At maximum load, for all cases three half-waves are noticeable. However, after the test damage, the mode for FEM_SOLID is relative to the experimental one. 

### 3.2. Compression at Elevated Temperature

Figure 12 presents the charts of compression force F_R_ vs. shortening Δ_T_ for experimental and numerical methods at temperature 50 °C. For each temperature, three samples were examined. These curves correspond to the third stage (after heating the column to reach adequate temperature), therefore at the beginning of chart, reaction forces for both methods could differ from each other (for Δ_T_ = 0). Taking into account results for 50 °C (Figure 12), the maximum load obtained in FEM is close to the maximum load observed in the experiment (EXP_mean_ = 13.54 kN). 

Moreover, in this case, obtained maximum loads are usually 60–70% greater in relation to the experimental result (See a comparison of the load-carrying capacities vs. temperature in the last Figure). As far as deformation shapes for temperature 50 °C are considered (Figure 13), at the end of second stage, all considered cases gave comparable views (one can see slight deflection of lateral walls). At the maximum load (F_Rmax_), three half-waves are visible, mostly. In the experiment, this phenomenon was not usually observed, maybe due to slightly different conditions of performed tests. Moreover, it could be caused by the imperfection of columns and the adjustment of models in the grooves of compressed plates. However, at the end of third stage, regions of high deformation are noticed in the lower part of column. This view was noticed also for FEM_SOLID. The discernible fluctuations of experimental curves might apparently be a result of local breaking of the adhesive due to reaching large strains, but this phenomenon was not possible to be verified during the test.

Figure 14 presents the curves of compression force F_R_ vs. shortening Δ_T_ for the experimental and numerical method at temperature 75 °C. The increase in reaction forces after the second stage is noticeable only for FEM calculations. The average load-carrying capacity taken from the experiment amounted to above 11 kN. Based on the numerical approach, the maximum loads (for all cases) usually gave higher values. Moreover, in contrast to numerical calculations, the reaction forces in the experiment during heating changed minimally or remained on the same level. Because the test lasted 1–2 h mostly, at a rise in temperature thermal deformations of columns could occur, which might change the shape of the column. This could be an explanation for the lack of distinct change in forces. The obtained maps of deformation for temperature 75 °C are shown in Figure 15. In general, they can be comparable with maps at 50 °C but a view of three-waves appearing along the column axis is not obvious (especially for maximum loads). The next diagram (Figure 16) depicts the characteristics of compression force F_R_ vs. shortening Δ_T_ at temperature 100 °C. The discrepancy between numerical and experimental curves is becoming greater. Similarly to results at temperature 75 °C, the reaction forces at the beginning of compression are unchangeable in the case of experiment, however the achieved peaks of load in FEM are slightly higher. The experimental average value of the load-carrying capacity amounted to above 10 kN. In the case of test at this temperature, obtained deformation maps at maximum load (point F_Rmax_) from FEM_SOLID and FEM_SHELL_2 turned out to be close to experimental maps (See Figure 17), but in the case of FEM_SHELL_2 the load-carrying capacity remains still too high in reference to average experimental maximum load which decreases with an increase in the temperature analysis. 

The next curves of compression attained at 125 °C, 150 °C, 175 °C and 200 °C (Figure 18, Figure 19, Figure 20 and Figure 21 respectively) indicate further decreases in the load-carrying capacities of columns due to the increase in temperature. 

The numerical results were illustrated only for FEM_SHELL because the full simulation based on FEM_SOLID with regard to the solution divergence was not able to be finished. The maps of the column deformation for higher temperatures turned out to be very comparable if these same numerical variants are considered for previous analysed temperatures (Figure 22, Figure 23, Figure 24 and Figure 25).

For higher temperatures applied in experiment, the regions of damage of columns are seen in the vicinity of the supports. In the case of FEM results (SHELL), this phenomenon is observed in the middle of column (FEM_SHELL_1) and on whole lateral walls for the second boundary condition (FEM_SHELL_2). Figure 26 illustrates the maximum loads vs. temperature for all studied cases. The average decrease in maximum load between ambient temperature and 200 °C experimentally amounted to 2.5 times. 

In the case of numerical calculations, a drop of the load-carrying capacity was noticed to be about two times. It can result in the fact that the separation of metals was possible only in the experiment (at higher temperatures and at greater strains glue ceased to ensure adequate connections) and this phenomenon surely might have happened as can be observed on the columns after finished tests (See columns after test—Figure 26 for temperatures 125–200 °C where distinct separations are visible). The estimated ratio of strength/weight for considered columns ranges from 94.63 kN/kg to 35.80 kN/kg for ambient temperature and for temperature 200 °C, respectively.

## 4. Summary

This paper concerns the study of compressed titanium-aluminium columns at elevated temperature. The experimental and numerical investigations were conducted to examine the behaviour and the load-carrying capacities of glued columns. Based on the results, the main conclusions can be stated:Mechanical properties of glued titanium-aluminium column depend on temperature therefore to validate numerical simulations in reference to experiment, mechanical parameters in a function of temperature should absolutely be taken into consideration.The load-carrying capacity of columns at temperature 200 °C decreased by 2.5 times in a comparison to maximum loads attained at ambient temperature. Apart from a decrease in the material properties at higher temperatures, meaningful drop could be caused additionally due to a separation of metals, because after test distinct dissection was noticed.Results of numerical model FEM_SOLID with similar boundary conditions (as in experiment) can be better referred to the experimental ones, whereas the discrepancies are additionally observed especially at higher temperature. Moreover, in contrast to FEM_SHELL simulation, FEM_SOLID calculations were more complex but they took significantly more time.The possible creeping of single metals (an intensification of this phenomenon grows with a growth of temperature) could occur while heating studied models which might have had an influence on the behaviour and the load-carrying capacity of columns.The process of separation metals at elevated temperatures through a long load action could play essential role in the change of the support conditions because apparently at big loads, sheets separately carried the compression forces.The maximum loads obtained numerically (FEM_SOLID, FEM_SHELL_1 with e = 1 mm) are slightly close to experimental ones, especially for lower temperatures. The bigger differences between experiment and simulation appear at higher temperatures. It should be mentioned that pursued analysis of the behaviour of thin-walled glued columns in thermal field is very complex because many parameters can matter. Firstly, analysed models could be imperfect based on a tolerance of thicknesses and dimensions, e.g., Secondly, mechanical properties of titanium, aluminum and glue are very temperature-dependent, whereas implemented one-directional tensile test curves (graded every 25 °C) into simulations were based on linear interpolation. Thirdly, the steel plates (upper and lower) used to keeping the columns are characterised by proper thermal expansion coefficient, whose value does not coincide with summary thermal expansion coefficient of components of column.Based on the results, the appearances of large deformations remaining in columns after tests are located mostly in the proximity of one of the supports.

## Figures and Tables

**Figure 1 materials-14-03013-f001:**
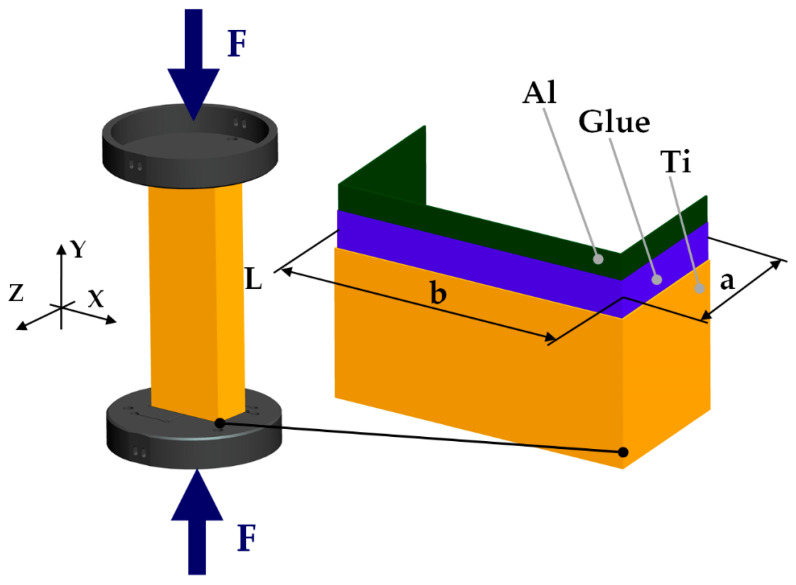
The object of the study.

**Figure 2 materials-14-03013-f002:**
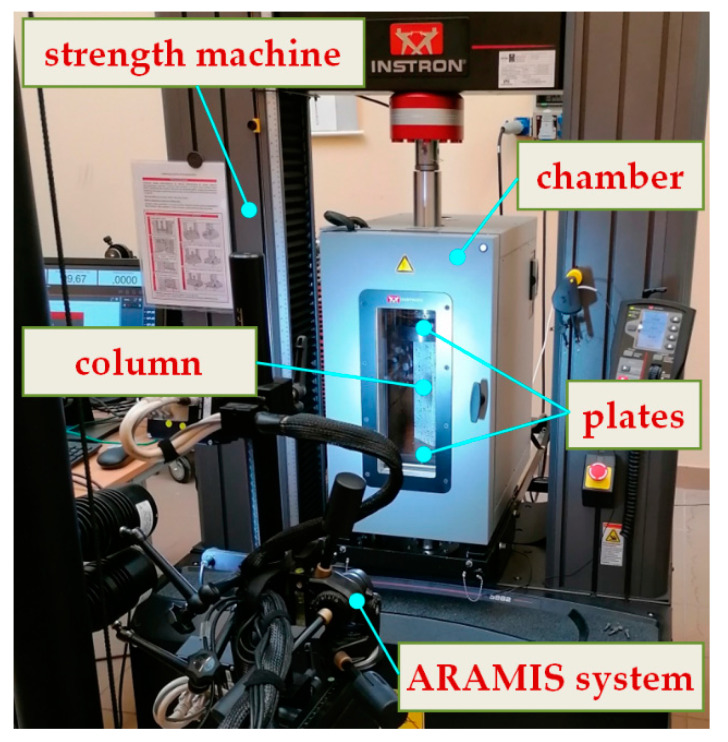
The testing stand equipped with ARAMIS^®^ system (Braunschweig, Germany).

**Figure 3 materials-14-03013-f003:**
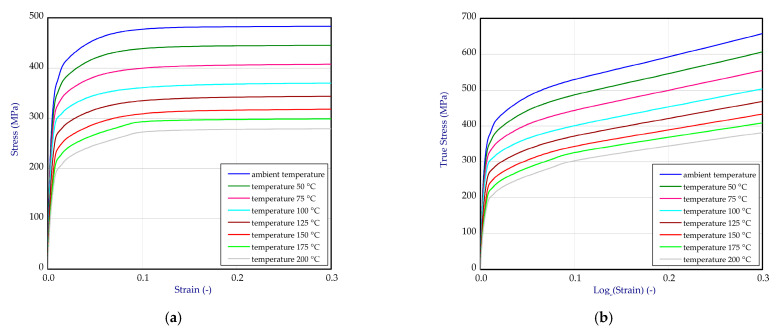
The tensile curves of titanium (**a**) and curves after transformation into true stress-logarithmic strain relation (**b**).

**Figure 4 materials-14-03013-f004:**
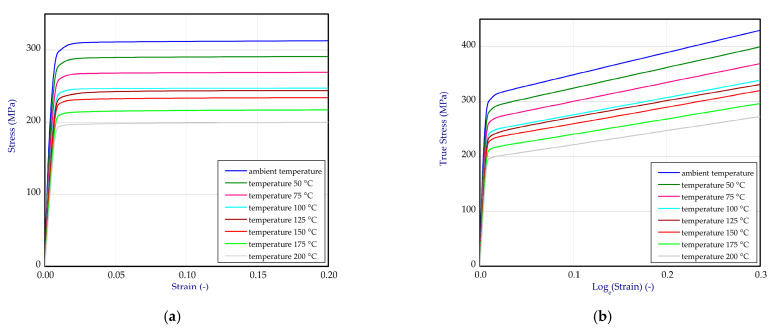
The tensile curves of aluminum alloy (**a**) and curves after transformation into true stress-logarithmic strain relation (**b**).

**Figure 5 materials-14-03013-f005:**
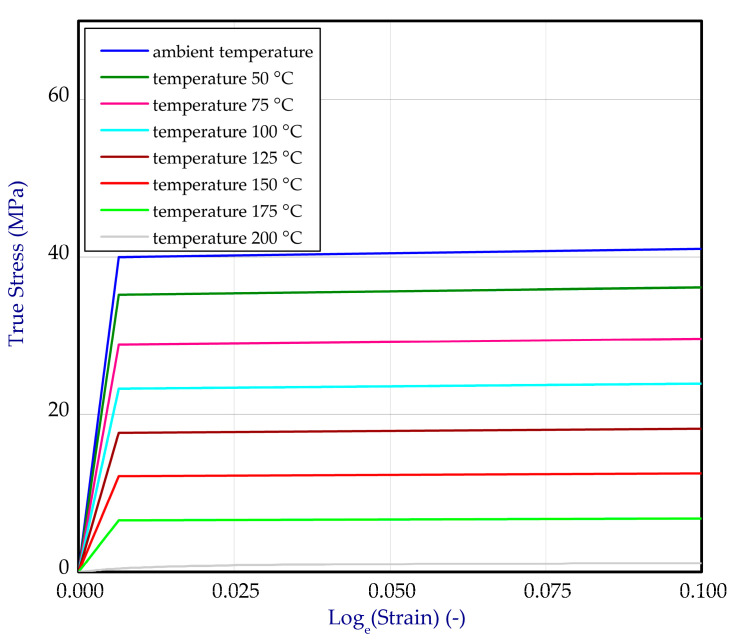
The tensile curves of true stress-logarithmic strain relation for adhesive.

**Figure 6 materials-14-03013-f006:**
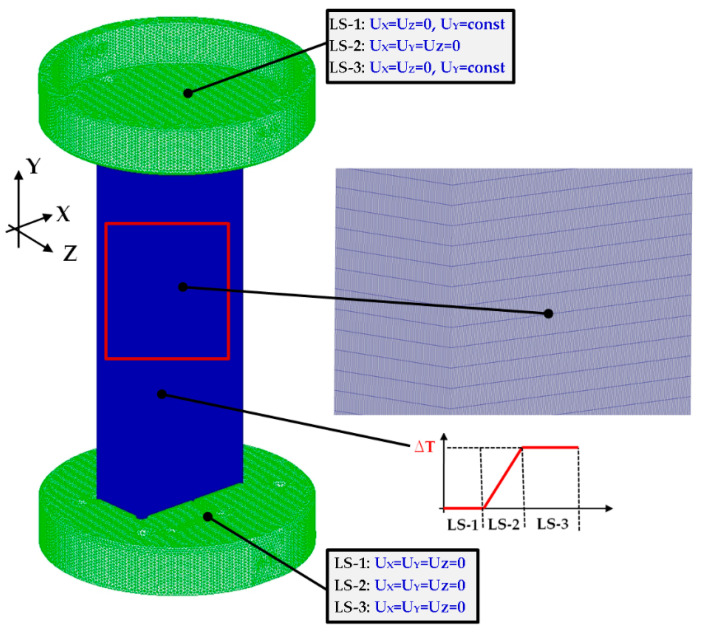
The discrete model FEM_SOLID with description of boundary conditions.

**Figure 7 materials-14-03013-f007:**
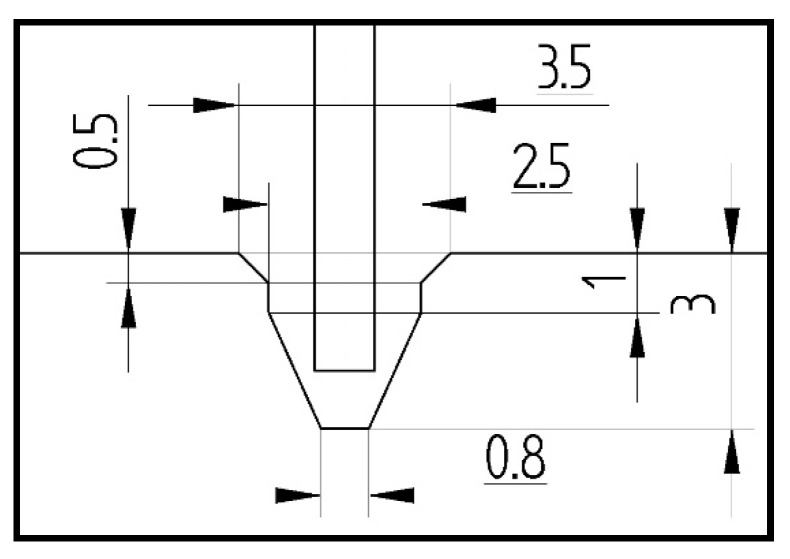
The support of column edges (shape of groove).

**Figure 8 materials-14-03013-f008:**
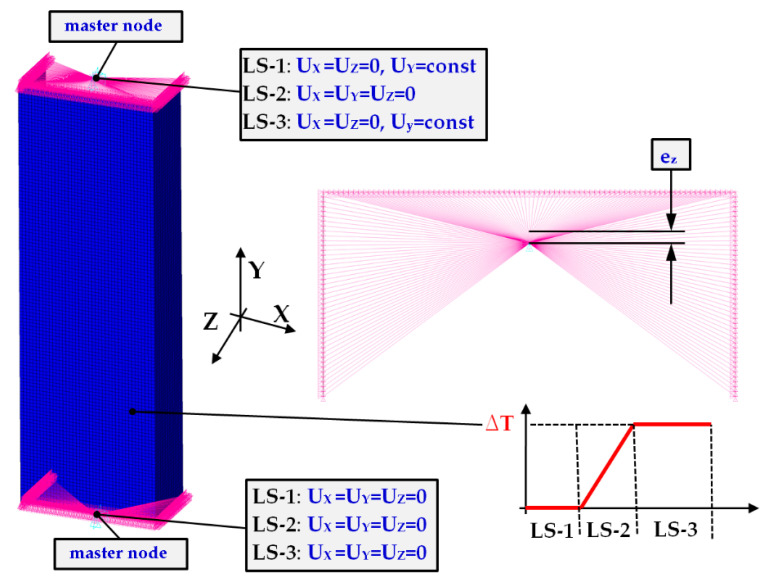
Boundary conditions of FEM_SHELL_1.

**Figure 9 materials-14-03013-f009:**
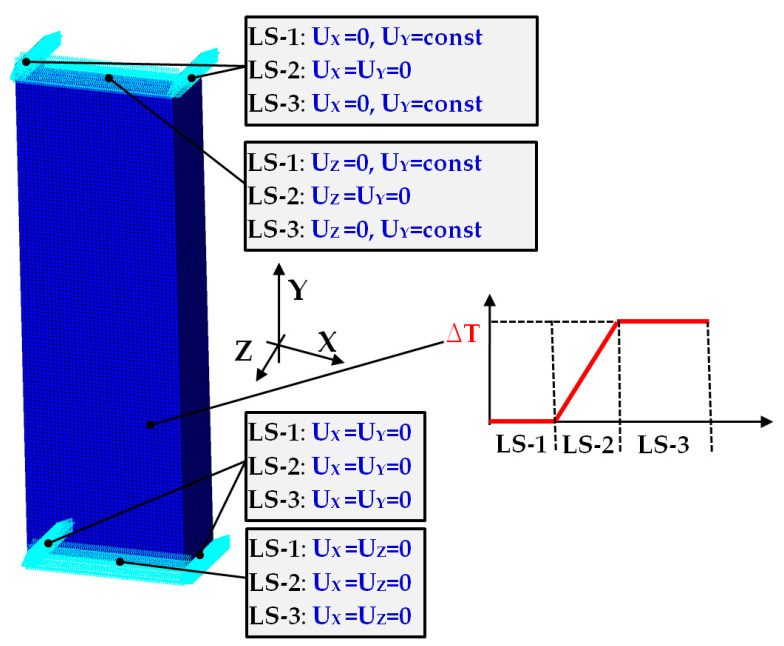
Boundary conditions of FEM_SHELL_2.

**Figure 10 materials-14-03013-f010:**
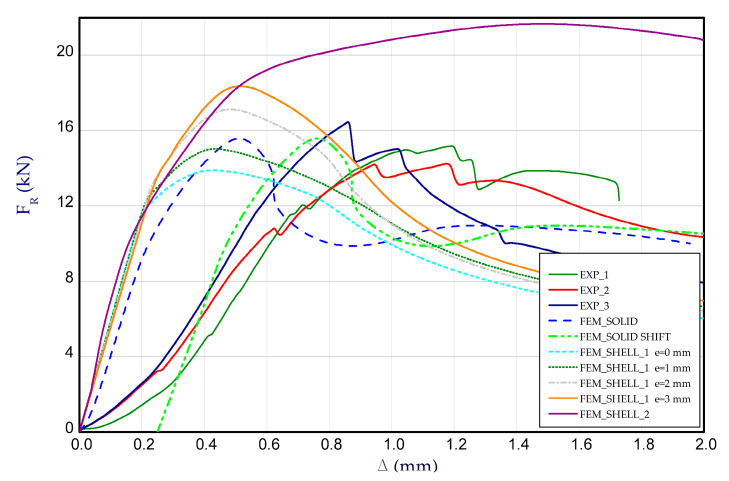
The compression curve (force vs. shortening) of column.

**Figure 11 materials-14-03013-f011:**
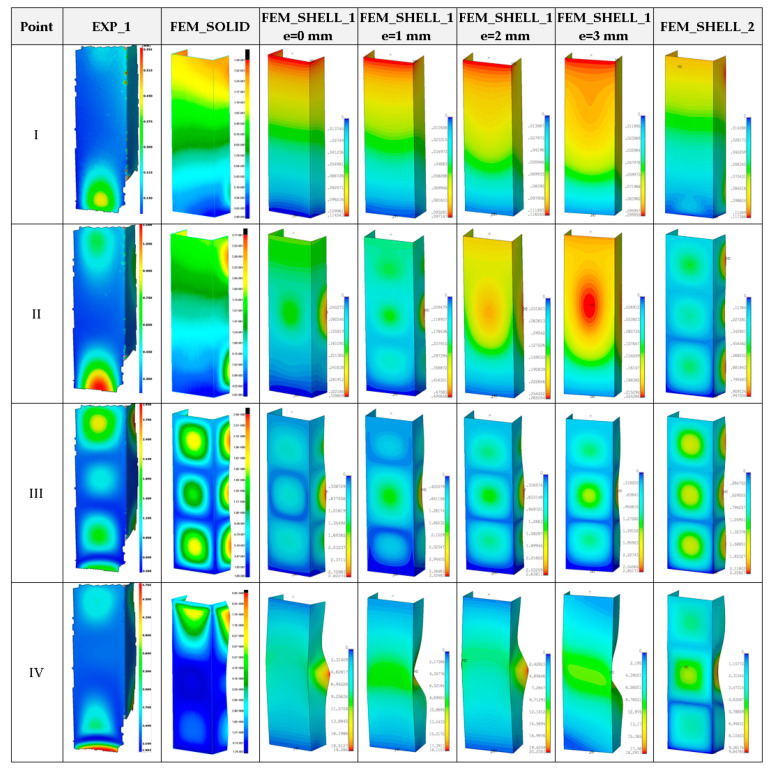
Deformation maps of columns at ambient temperature.

**Figure 12 materials-14-03013-f012:**
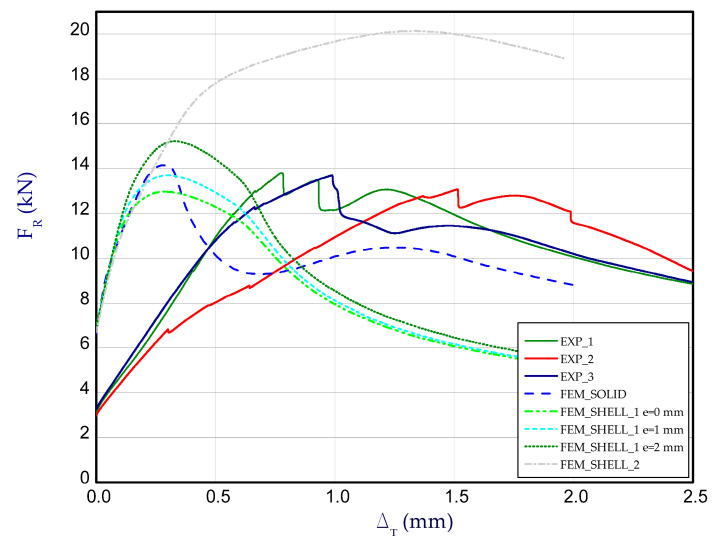
The compression curves of column at 50 °C.

**Figure 13 materials-14-03013-f013:**
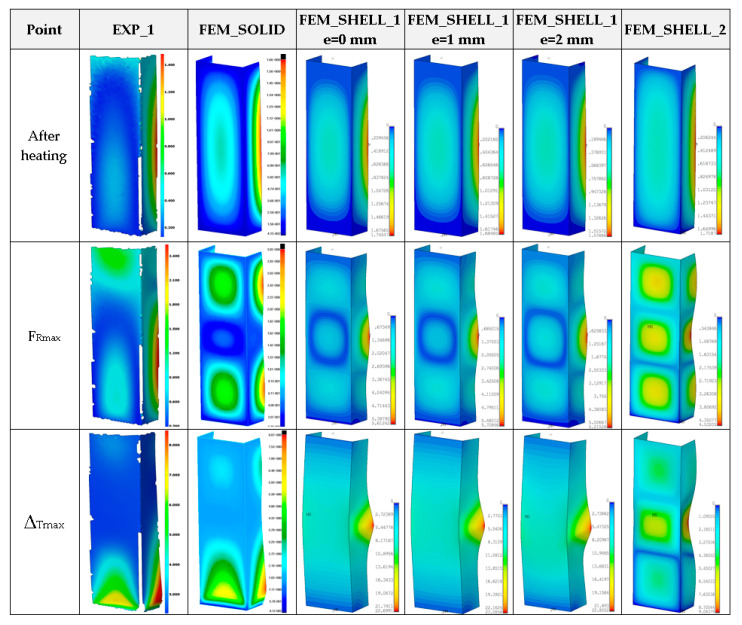
Deformation maps of columns at 50 °C.

**Figure 14 materials-14-03013-f014:**
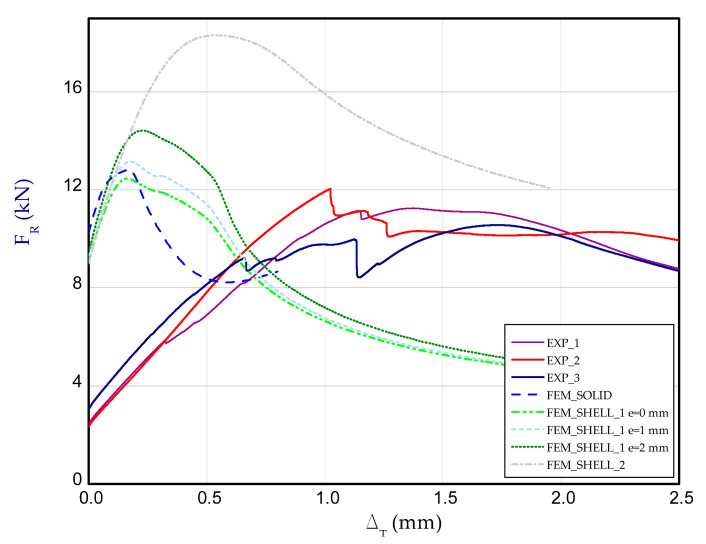
The compression curves of column at 75 °C.

**Figure 15 materials-14-03013-f015:**
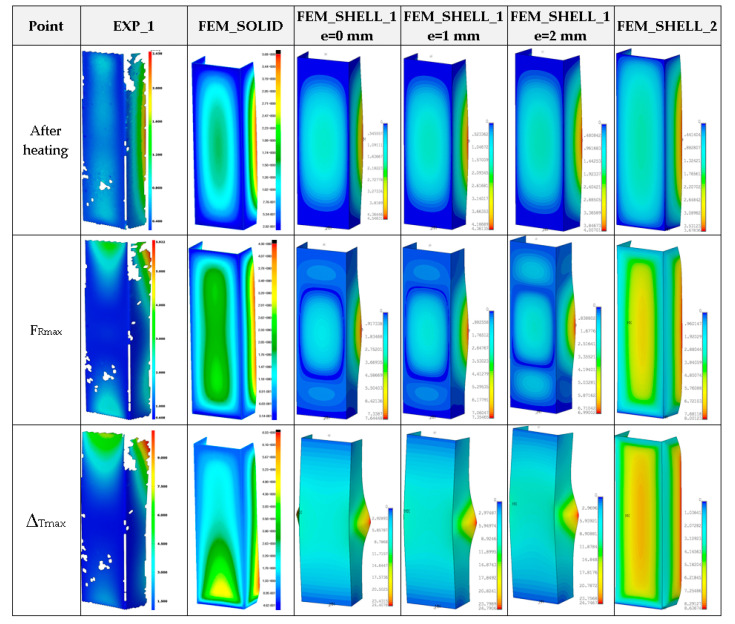
Deformation maps of columns at 75 °C.

**Figure 16 materials-14-03013-f016:**
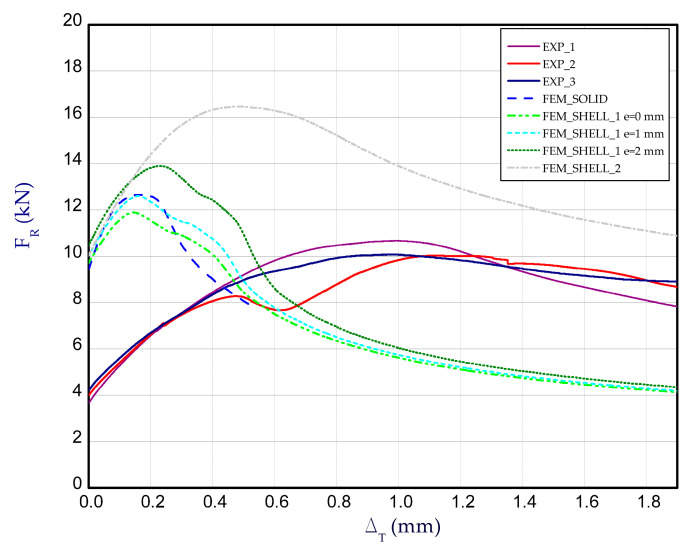
The compression curves of column at 100 °C.

**Figure 17 materials-14-03013-f017:**
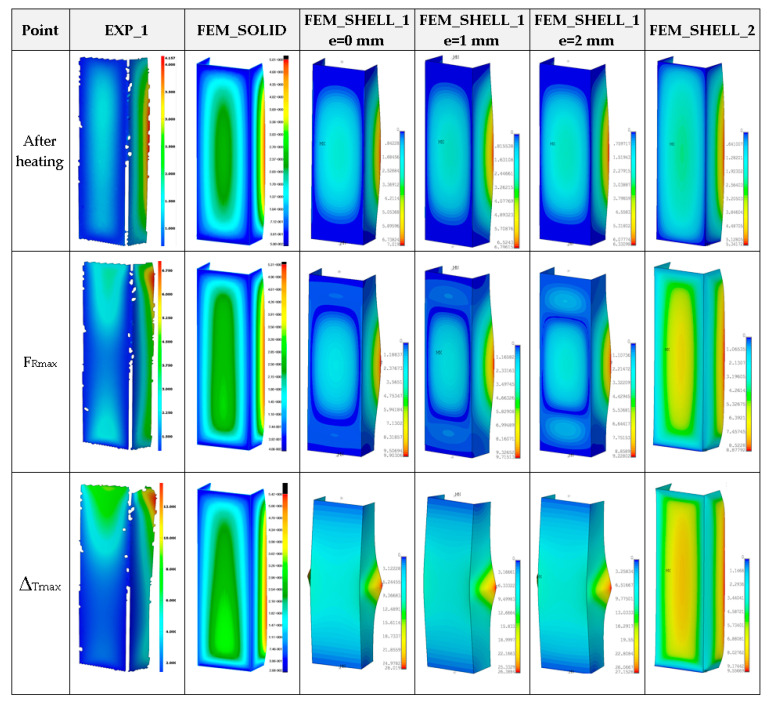
Deformation maps of columns at 100 °C.

**Figure 18 materials-14-03013-f018:**
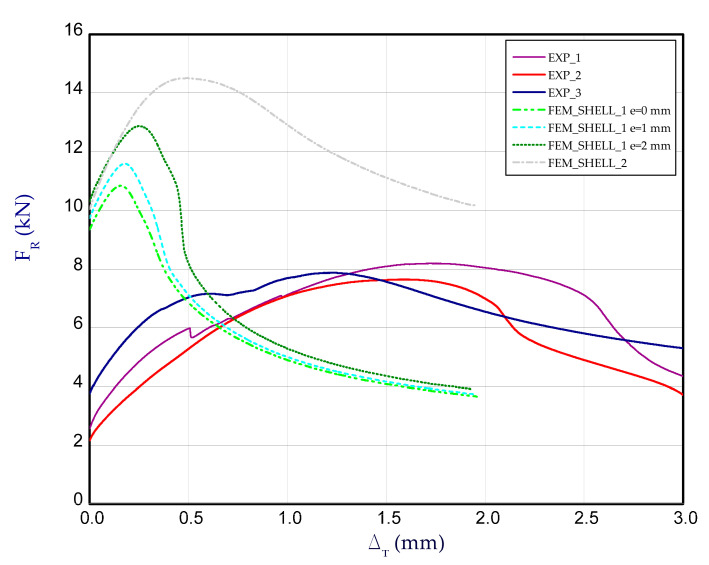
The compression curves of column at 125 °C.

**Figure 19 materials-14-03013-f019:**
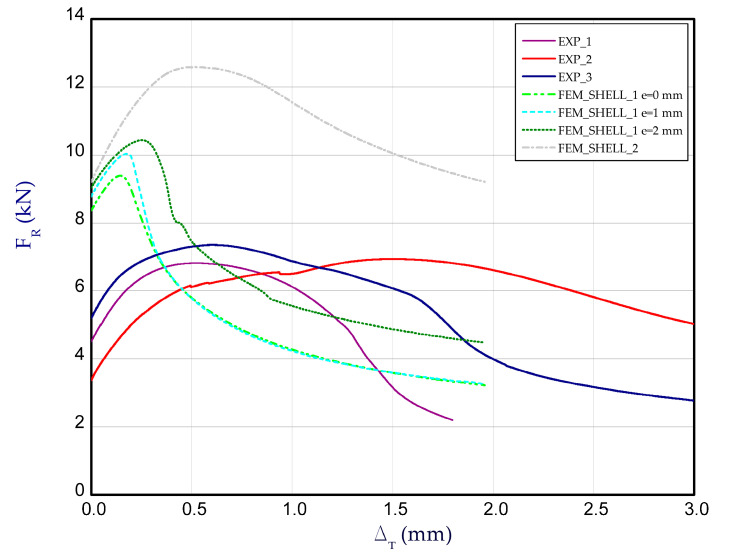
The compression curves of column at 150 °C.

**Figure 20 materials-14-03013-f020:**
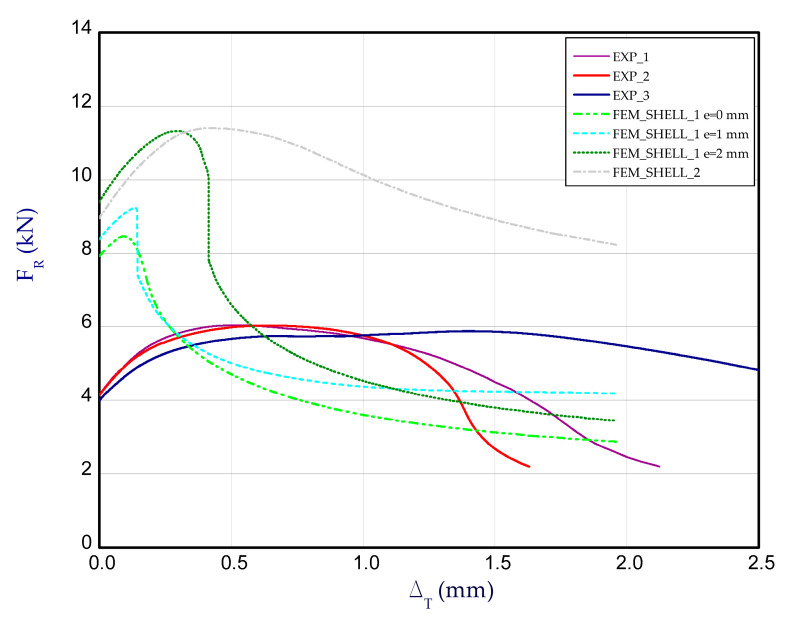
The compression curves of column at 175 °C.

**Figure 21 materials-14-03013-f021:**
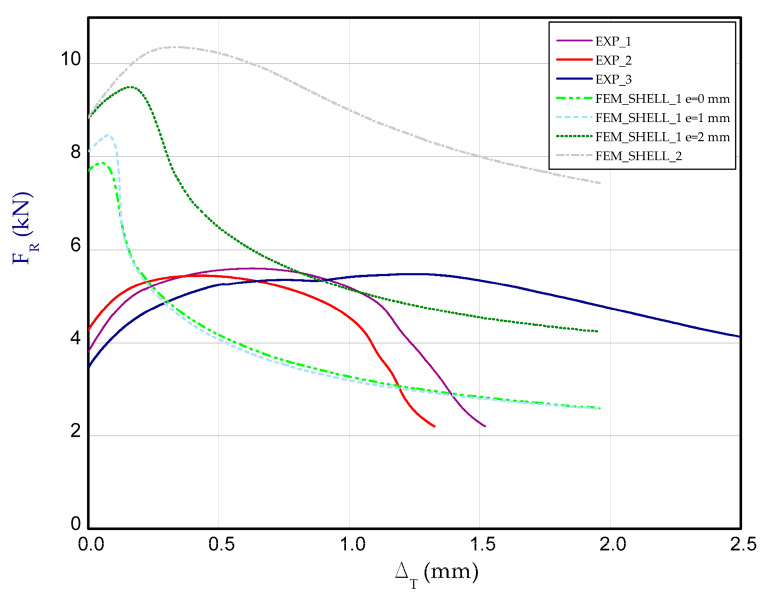
The compression curves of column at 200 °C.

**Figure 22 materials-14-03013-f022:**
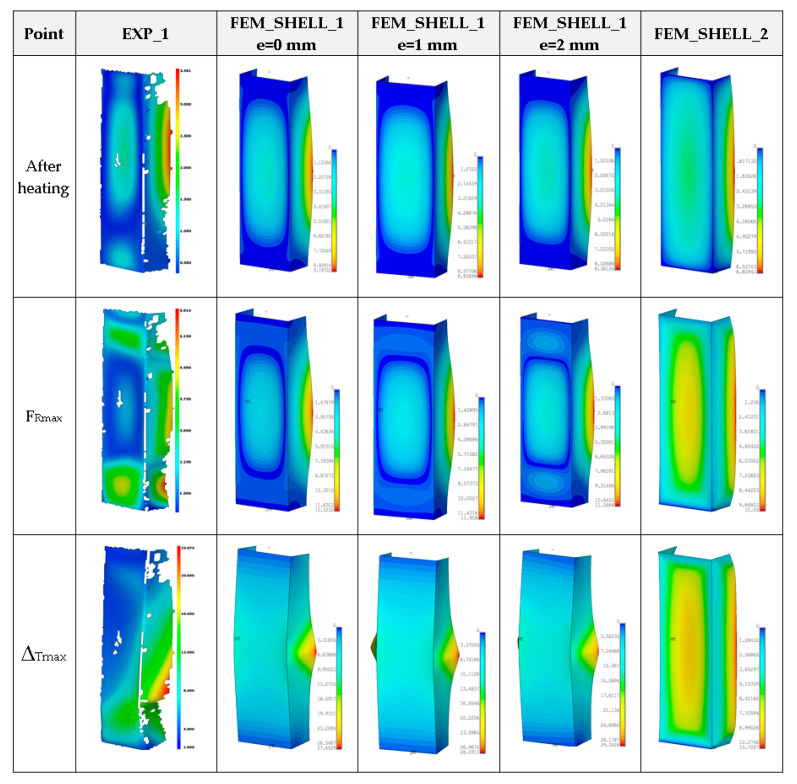
Deformation maps of columns at 125 °C.

**Figure 23 materials-14-03013-f023:**
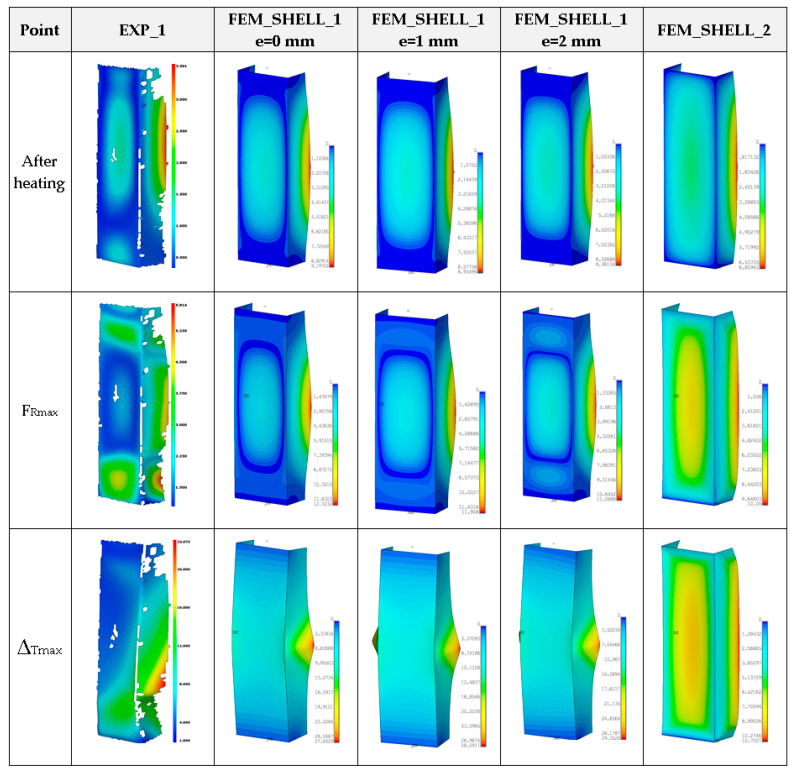
The compression curves of column at 150 °C.

**Figure 24 materials-14-03013-f024:**
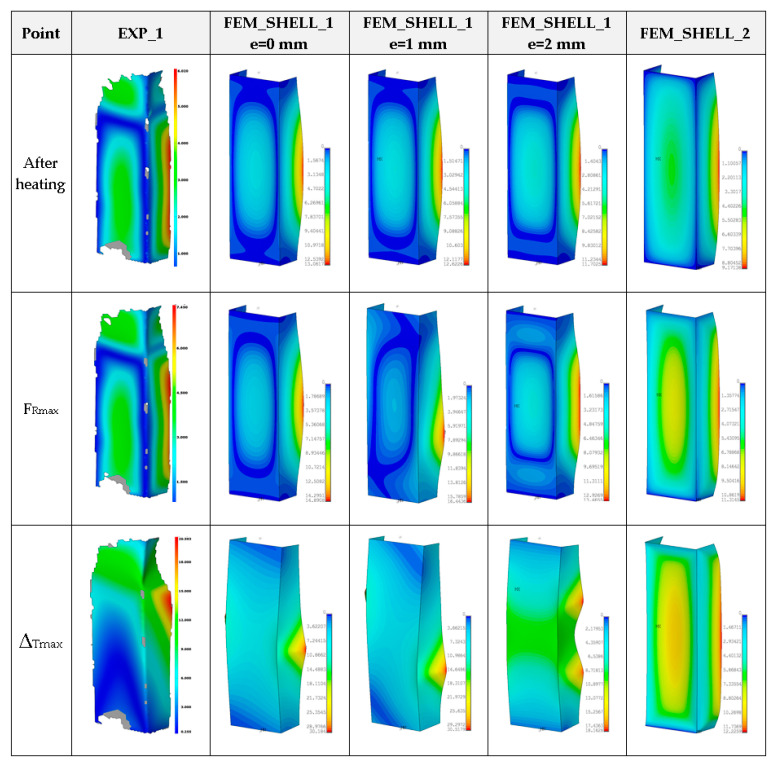
The compression curves of column at 175 °C.

**Figure 25 materials-14-03013-f025:**
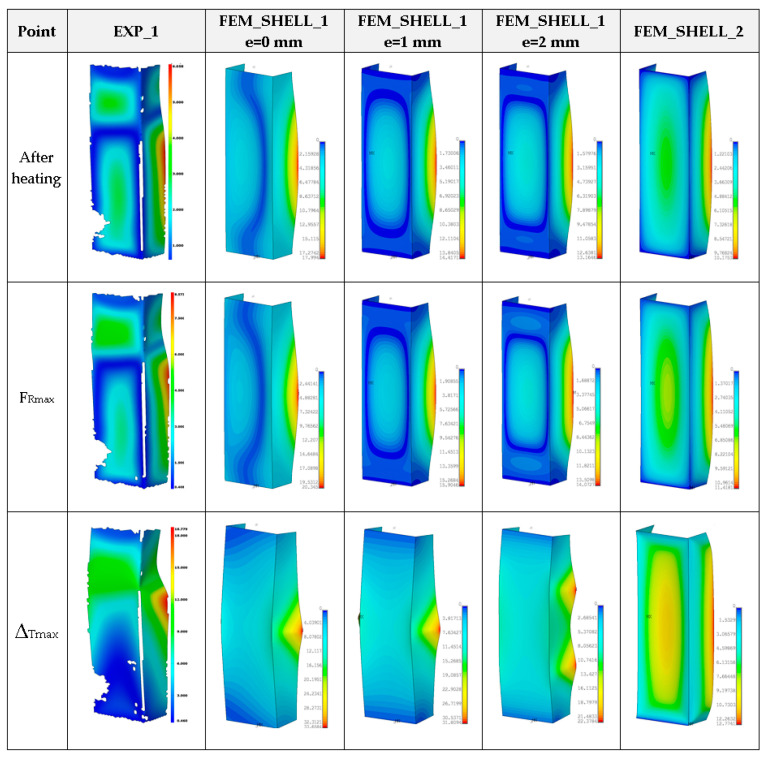
The compression curves of column at 200 °C.

**Figure 26 materials-14-03013-f026:**
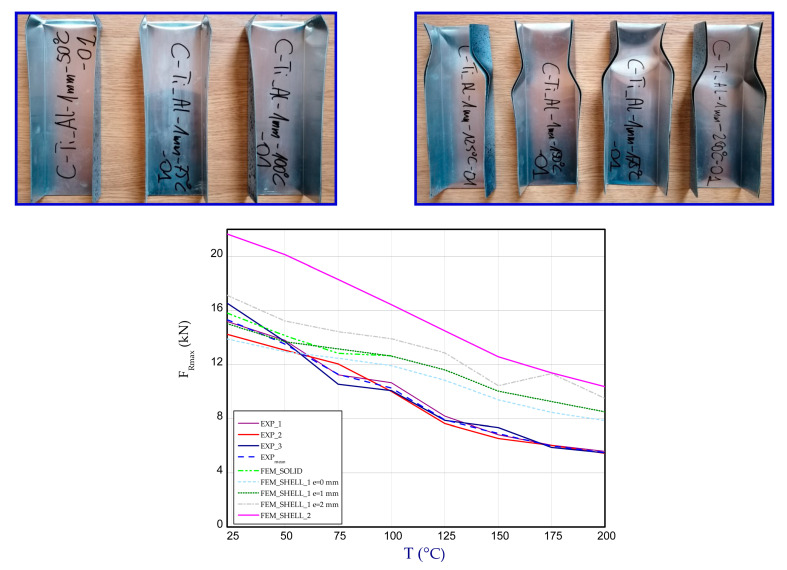
Peaks of loads for all cases with pictures of damaged columns after test (from the left side to the right side for 50 °C, 75 °C, 100 °C, 125 °C, 150 °C, 175 °C, 200 °C, respectively).

**Table 1 materials-14-03013-t001:** The maximum reaction forces obtained at ambient temperature in (kN).

EXP_1	EXP_2	EXP_3	EXP_mean_	FEM_SOLID	FEM_SHELL_1 e = 0 mm	FEM_SHELL_1 e = 1 mm	FEM_SHELL_1 e = 2 mm	FEM_SHELL_1 e = 3 mm	FEM_SHELL_2
15.20	14.25	16.55	15.33	15.82	13.90	15.04	17.14	18.36	21.66

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
