# Peer review of "Determination of Load-Carrying Capacity of C-Profile Glued Ti-Al Column under Temperature Environment"

_materials, 2021, doi:10.3390/ma14113013_

Round 1

Reviewer 1 Report

Good work, in plain, about compressed titanium-aluminium columns at elevated temperature. The experimental and numerical investigations were conducted to examine the behaviour and the load-carrying capacities of glued columns. Modelling and experimental test are offered.

buckling lis always welcom in research

 There are several inputs from your country, that is OK, however some missing gaps are detected:

  • Digital Image 10 Correlation Aramis® System (DICAS): please expalin a Little more is this is comemercial softwsre and if you did an adpatation to your case.
  • Araldite® AW 4804/Hardener HW 4804: can you give more details?
  • Border effects are key in many buckling or fatigue or tensile problems, see all the discussion in the Society of Experimental Mechanics after the publication of Alternatives for specimen manufacturing in tensile testing of steel plates, Experimental Techniques 40 (6), 1555-1565 in this case in steels and Inconel, but you can have similar concerns in Ti. Please include the idea. Border effects are very affecting results

Thin walls is becoming on hot topic in several applications. Can you give the ratio strength/weight in some samples, this is key, and in some stiffeners in large dies as well.

Make a better introduction with the missed border effect and paper could reach a good quality standard

Author Response

Dear Reviewer,

I’d like to thank for your review. The added/changed text was highlighted on “yellow”. In case of acceptance before publishing, the English in manuscript will be verified once again by Proofreader.

With best Regards,

L. Czechowski

Reviewer 2 Report

The paper presents the outcome of an analysis on the behavior of glued Ti-Al column under compression at high temperatures. Such analysis combines compression tests and numerical simulations exploiting the Finite Element Method and two different discrete models. Numerical simulations based on Green–Lagrange equations have been developed aimed at testing large deflections and strains values, by highlighting how temperature affects the behavior of compressed C-profile Ti-Al columns.

The topic of the paper is quite interesting, although very specific, and completely aligned to the aims and scope of the Journal. The adopted investigation methodology presents limited elements of novelty, but the application, i.e. the considered material and components, is not widely investigated. Therefore, the paper can be a good example of coupled experimental and simulation-based investigation, which can be of didactical interest for students and practitioners and this is its main added value.

The introduction provides a quite complete overview of the state of the art on the concerned topics, but the purpose of the paper should be better emphasized through a more complex dedicated statement. The author seems not to be sure of the innovative contents of its own paper, as the following sentence is found. “… the present study SEEMS  to be novelty because stability of glued profiles as FML…”. Why this?

The experimental setup and the considered material are clearly described in Section 2. The two adopted Finite Element Models are also described in Section 2, but the selected value for number of elements is not fully justified and unclear. This aspect needs to be more elaborated by the author.

The results are overall extensively described and deeply discussed in Section 4, and the achieved conclusions are well supported by the provided data.

The English language needs some revisions, as there are a number of rough and not fully correct expressions. See the following exemplar and not exhaustive list:

  • There is a lack of commas and other punctuation throughout the whole paper;
  • Page 2, line 65: the sentence “…the present study seems to be novelty because stability of glued profiles as FML…” should be replaced with something like “…the present study is novel, as the stability of glued profiles as FML…”;
  • Page 2, lines 81-82: The following sentence “… strengths of metals were different what essentially could influence on behaviour of columns.” Should be replaced by: “…strength values of the considered metals were different, and this fact can substantially affect the columns behavior”;
  • Page 18, lines 347-348: the sentence “…the appearance of large deformations in the columns after finished tests occurred mostly in a vicinity of the one of the support” should be replaced with “…large deformations remaining after the conclusions of the tests are located mostly in proximity of one of the supports” or something similar.

I suggest the author to exploit the support of a professional English proofreader.

As further minor formal remarks, the use of acronyms should be avoided in the abstract, and captions should be placed in the same page of the corresponding figure.

Author Response

(The authors gave the same response as above.)
